# A Putative Effector Pst-18220, from *Puccinia striiformis* f. sp. *tritici*, Participates in Rust Pathogenicity and Plant Defense Suppression

**DOI:** 10.3390/biom14091092

**Published:** 2024-08-31

**Authors:** Mengfan Tian, Zhen Zhang, Xiaorui Bi, Yan Xue, Jiahui Zhou, Bo Yuan, Zhaozhong Feng, Lianwei Li, Junjuan Wang

**Affiliations:** School of Life Science, Jiangsu Normal University, Xuzhou 221116, China; 2020221659@jsnu.edu.cn (M.T.); 3020222675@jsnu.edu.cn (Z.Z.); 3020222598@jsnu.edu.cn (X.B.); 3020222696@jsnu.edu.cn (Y.X.); 3020222905@jsnu.eud.cn (J.Z.); boyuan@jsnu.edu.cn (B.Y.); fzz2012@jsnu.edu.cn (Z.F.); lwli@jsnu.edu.cn (L.L.)

**Keywords:** wheat stripe rust, host-induced gene silence (HIGS), plant immunity

## Abstract

Stripe rust, caused by *Puccinia striiformis* f. sp. *tritici* (*Pst*), stands out as one of the most devastating epidemics impacting wheat production worldwide. Resistant wheat varieties had swiftly been overcome due to the emergence of new virulent *Pst* strains. Effectors secreted by *Pst* interfere with plant immunity, and verification of their biological function is extremely important for controlling wheat stripe rust. In this study, we identified an effector, Pst-18220, from *Puccinia striiformis* f. sp. *tritici* (*Pst*), which was induced during the early infection stage of *Pst*. Silencing the expression of Pst-18220 through virus-mediated host-induced gene silencing (HIGS) resulted in a decreased number of rust pustules. In *Nicotiana benthamiana*, it significantly suppressed cell death induced by *Pseudomonas syringae* pv. *tomato* (*Pto*) DC3000. In Arabidopsis, plants with stable overexpression of Pst-18220 showed increased susceptibility to *Pto* DC3000, accompanied by a decrease in the expression level of pattern-triggered immunity (PTI)/effector-triggered immunity (ETI)-related genes, namely, AtPCRK1, AtPCRK2, and AtBIK1. These results emphasize the significant role of the *Pst* candidate effector, *Pst*-18220, in rust pathogenicity and the suppression of plant defense mechanisms. This broadens our understanding of effectors without any known motif.

## 1. Introduction

Stripe rust, caused by *Puccinia striiformis* f. sp. *tritici* (*Pst*), stands out as one of the most devastating epidemics impacting wheat production globally [1]. The economic toll of wheat stripe rust outbreaks is staggering, ranging from USD 4 billion to USD 5 billion annually [2]. Numerous resistant wheat varieties are currently employed in agricultural production to combat *Pst*. However, this control strategy has swiftly become ineffective due to the emergence of new virulent *Pst* strains exhibiting significantly heightened aggressiveness, which override these resistances [3]. Also, chemical pesticides are widely used in control of this disease, but this strategy causes environmental pollution and increases strain resistance. Consequently, unveiling the molecular underpinnings of *Pst* infection assumes paramount importance in the quest to devise novel, enduring strategies for controlling this fungal disease.

During plant-pathogen interactions, plants resist pathogen attack due to their innate immune system. When pathogens land on the surface of the host, plant plasma membrane-resident pattern recognition receptors (PRRs), including receptor proteins (RPs), can sense conserved pathogen-associated molecular patterns (PAMPs), leading to the rapid initiation of basal immune responses, known as pattern-triggered immunity (PTI) [4]. Conversely, to successfully colonize plants, pathogens deliver a large number of virulence proteins, called effectors, into host cells [5]. Simultaneously, plants have evolved immune receptors to directly or indirectly recognize specific effectors, thereby activating effector-triggered immunity (ETI) [6]. Recently, increasing evidence suggests an intricate interaction between PTI and ETI, resulting in a robust plant defense against pathogen infections [4,7]. The activation of PTI and/or ETI involves a series of signaling cascades and responses, such as rapid Ca^2+^ flux, a transient burst of reactive oxygen species (ROS), hypersensitive cell death response, stomatal closure, callose deposition, and hormone dynamics [4]. 

Effectors that interfere with plant immunity typically contain specific motifs or exhibit richness in particular amino acids (AA). Certain effectors from *Magnaporthe oryzae* (*M. oryzae*), for example, are abundant in glycine/serine, participating in the regulation of various antioxidant enzymes. This regulation leads to a reduction in reactive oxygen species (ROS) levels within the host, thereby suppressing the host’s immune response [8,9]. A similar scenario is observed in *Puccinia striiformis* f. sp. *tritici* (*Pst*). In *Pst*, the effector PstGSRE1 contains a glycine-serine-rich motif (m9) and interacts with a wheat transcription factor, TaLOL2. This interaction suppresses ROS-induced cell death, ultimately leading to the inhibition of host immunity [10]. Additionally, Pst27791, a serine-rich effector, can interact with the wheat rapidly accelerated fibrosarcoma (Raf)-like kinase TaRaf46, suppressing ROS accumulation [11]. Two cysteine-rich effectors from *Pst*, PSTG_14695, and PstSCR1, interfere with the suppression of the plant defense response [12,13]. However, not all effectors in *Pst* exhibit known sequence domains. For instance, PSTha5a23 lacks any known sequence motifs and is localized to the cytosol. This effector significantly enhances the virulence of *Pst* and suppresses pathogen-associated molecular PTI-related callose deposition [14]. Another effector, PstSIE1, also lacking any known sequence motifs, disrupts the formation of the chaperone complex TaRAR1–TaSGT1 that plays a crucial role in plant immunity modulation. This disruption results in the promotion of pathogenesis [15]. These findings suggest that effectors without known motifs may possess specific structural characteristics that promote *Pst* infection.

As an obligate biotrophic pathogen, *Pst* cannot be cultured in vitro and must extract nutrients from living plant tissues [16], hindering research progress in uncovering the detailed biology of *Pst* infection. Recently, a wide range of effector repertoires in *Pst* has been identified, benefiting from a series of genome sequence datasets [17,18,19,20]. Therefore, their roles in interfering with plant defense responses still need further studies.

The objective of this study is to investigate the function of a putative effector of *Pst*, Pst-18220, providing detailed evidence for Pst-18220 affecting *Pst* pathogenicity and plant immunity. This study lays a foundation for our understanding of *Pst* effectors without any known motif, providing a theoretical foundation for us to control wheat stripe rust.

## 2. Materials and Methods

We used a flowchart (Figure 1) to help readers better understand our experimental process.

### 2.1. Plant Materials and Rust Inoculation

The wheat (*Triticum aestivum*) cultivars Mingxian 169 and *Pst* CYR32 were utilized in this study. Wheat plants were cultivated in a glasshouse; specifically, seeds [6,7,8]. were cultivated in a plastic pot (10 × 10 × 10 cm^3^) filled with a potting mixture under rust-free conditions. At the two-leaf stage of the seedlings (about 14 days), the first leaves were uniformly brushed with a mixture of *Pst* urediniospores and sterile water at a ratio of approximately 1:10–15 (*v*/*v*). Then, seedlings were placed in a dew chamber in the dark for 24 h (temperature, 10 °C; relative humidity, 90–100%) and subsequently transferred to a growth chamber (day:night, 16 °C:12 °C, 16 h:8 h) with a relative humidity of 60–80%. Leaf tissues were sampled at 0, 6, 12, 24, 48, 72, 120, 168, 214, and 264 h post inoculation (hpi). Three biological replicates were employed for each assay.

The *N. benthamiana* employed for cell death was also grown in a glasshouse (16:8 h, 24:20 °C, day:night). *Arabidopsis thaliana* (L.) Heynh. Col-0 plants underwent transformation and were used to test the interference of Pst-18220 on plant immunity.

### 2.2. Cloning and Phylogenetic Analyses of Pst-18220

Based on the candidate effector 18220 from isolate 08/21 (location: UK, occurrence: 2008) of *Pst* [17], a BLAST was performed on the genome of *Pst* CYR32, and Pst-18220 was obtained through homology-based cloning. The PCR product of Pst-18220 was subsequently cloned into the PMD18-T vector (TaKaRa, Tokyo, Japan) for sequencing. The primers used for cloning this gene are presented in Appendix A. Signal peptide prediction was conducted using SignalP-5.0 software (https://services.healthtech.dtu.dk/service.php?SignalP-5.0, accessed on 10 March 2024), while nuclear localization signal prediction was carried out using PSORT v. 3.0 (http://www.genscript.com/psort.html, accessed on 10 March 2024). The analysis of conserved protein motifs was performed using the MEME online website (https://meme-suite.org/meme/, accessed on 10 March 2024).

MEGA 11 software was used for phylogenetic tree analysis using a bootstrap test of phylogeny with a minimum evolution test and default parameters of 1000 replications. ClustalX v. 2.1 (http://www.clustal.org/download/current/, accessed on 1 March 2024) software was employed to conduct multiple comparisons of protein sequences. The protein sequences utilized for constructing the phylogenetic tree were searched using BLASTP in NCBI (Protein BLAST: search protein databases using a protein query (nih.gov, accessed on 10 January 2024), and their detailed information is presented in Appendix A.

### 2.3. RNA Extraction and cDNA Synthesis

Total RNA was extracted from the leaves of *Arabidopsis thaliana* infected with *Pto* DC3000 and wheat infected with *Pst* CYR32. Genomic DNA contamination was eliminated through DNase I treatment (Thermo Fisher, Waltham, MA, USA), and the resulting clean mRNA was converted into cDNA using the PrimeScriptVR RT Reagent kit (TaKaRa, Japan).

The qRT-PCR reactions were performed using FastSYBR (Sangon Biotech, Shanghai, China) mixture and conducted in an ABI7500 instrument (ABI). The primers are provided in Appendix A. In all of the experiments, data were collected from three independent biological replicates, each consisting of at least three reactions, and negative controls with sterilized ddH_2_O were assessed to ensure the absence of contamination. To test the relative expression level of *Pst-18220* in rust fungus, the elongation factors *Pst-EF1* were used as reference genes, and *At-actin* was utilized as reference genes to assess the relative expression of *AtPCRK1*, *AtPCRK2,* and *AtBIK1* in *A. thaliana*. The relative expression of all genes was calculated using the 2^–ΔΔCt^ method [21]. The qRT-PCR protocol included an initial step of 95 °C for 3 min, followed by 40 cycles of 95 °C for 20 s, 60 °C for 30 s, and 72 °C for 30 s.

### 2.4. Construction of Plasmids

Plasmids employed for silencing Pst-18220 were constructed following previously established procedure [22]. To ensure the specificity of gene silencing, BLASTn and BLASTx searches were conducted against the National Center for Biotechnology Information (NCBI) database. Two cDNA fragments (223 bp and 250 bp) originating from Pst-18220 were amplified with a pair of primers (Appendix A) and used to construct the γRNA-based derivative plasmids from barley stripe mosaic virus (BSMV), BSMV:Pst-18220-1/BSMV:Pst-18220-2 recombinant plasmids. Wheat phytoene desaturase (PDS) gene (TaPDS) was then replaced with this fragment in BSMV:γ-PDS. For the generation of transgenic *A. thaliana* and the assessment of cell death in *N. benthamiana*, PCR fragments derived from the complete coding sequence of Pst-18220 with BamHI (5′) and XbaI (3′) were incorporated into the PHB vector using homologous recombination facilitated by the one-step cloning kit (Vazyme, Nanjing, China).

### 2.5. HIGS-Mediated Pst-18220 Silencing

The vectors utilized for gene silencing were linearized and transcribed into infectious BSMV RNAs using an in vitro high-yield capped RNA transcription kit (Ambion® mMESSAGE mMACHINE®, Boston, USA). The BSMV inoculum consists of 2.5 μL of α, β, and genetically modified γ transcripts with 42.5 μL of FES (viral inoculation buffer, 1 g sodium pyrophosphate, 1 g diatomaceous earth, 0.75 g glycine, 1 g bentonite, and 1.05 g dipotassium dihydrogen phosphate are added to 100 mL of 0.1% DEPC water) [23] and was inoculated onto the second leaves of wheat plants at the two-leaf stage by gently rubbing the leave surface using a gloved finger [24]. The vectors BSMV:00 and BSMV:TaPDS served as the negative and positive controls, respectively. Following inoculation, the wheat plants were cultivated in a growth chamber under conditions of 25 °C (16:8 h, day:night). Nine days post-virus inoculation, the fourth leaves of wheat were inoculated with fresh urediniospores of *Pst* CYR32 as previously described. Subsequently, they were transferred into an incubator (10 °C, 24 h, dark) and then to a growth chamber (16:8 h, 16 °C:12 °C, day:night). After 14 days of *Pst* inoculation, *Pst* sporulation, and stripe rust symptoms on the fourth leaves were examined and documented through photography.

### 2.6. Histological Observations of Pst-18220-Knockdown Wheat Plants

Wheat leaves from HIGS-mediated Pst-18220 silencing were subsequently subjected to *Pst* inoculation. At 120 h and 14 days post *Pst* inoculation, wheat leaves were sampled to assess changes in *Pst* development. Staining procedures were conducted as described by Wang et al. [25]. After fading, fixation, and dyeing (0.1% Calcofluor White M2R), cleared leaf segments were assessed to determine the hyphal length and number of haustorial mother cells from 120 h and the number of uredinium from 14 days of *Pst* infection. Hyphal length was measured from the substomatal vesicle to the apex of the longest hypha. The number of uredinium per leaf was tested with a magnifying glass. About 20–50 infection sites on 10–20 wheat leaf segments (length, 1.5 cm) from 10–15 randomly selected wheat plants were examined. All of the microscopic observations were carried out using a Leica DFC550 microscope (Frankfurt, Germany).

### 2.7. Transient Expression in N. benthamiana

Recombinant vectors PHB-Pst-18220 and PHB-Pst-18220^Δsp^ were constructed and subsequently transformed into *A. tumefaciens* GV3101 for transient expression. The transformed strain was shaken at 28 °C for 16 h and then centrifuged at 4000 g for 20 min. The Agrobacterium pellets were suspended in permeation buffer (10 mM 2,4-morpholinoethanesulfonic acid [MES], 10 mM MgCl_2_, and 150 nM acetosyringone), and the OD_600nm_ was adjusted to 0.8. The resuspension was injected into the leaves of 5-week-old *N. benthamiana* using a syringe and kept at 25 °C (16 h:8 h, day:night) for 24 h in a chamber. In parallel, the injection of MgCl_2_ (10 mM) and PHB-GFP served as the negative control. After 24 h of transformed Agrobacterium injection, the bacterial *Pto*DC3000 (OD_600nm_ = 0.01) was injected at the same site where the transformed strain was primarily injected as well as the negative control. After 48 h of *Pto*DC3000 inoculation, sites exhibiting necrosis were considered as cell death, and the phenotypes were photographed.

### 2.8. Constitutive Overexpression in A. thaliana

A. tumefaciens Smith and Townsend GV3101, following the protocol detailed by Clough and Bent [26], was used to produce overexpression of PHB-Pst-18220/PHB-Pst-18220^Δsp^ in *A. thaliana*. The vector construction was the same as that used in the transient expression in *N. benthamiana*. The successful transgenic plants were verified on half-strength MS selection plates containing 50 μg mL^−1^ hygromycin and 100 μg mL^−1^ timentin. These plants were then grown in a growth chamber with a light cycle of 16 h at 22 °C during the day and 8 h at 18 °C during the night. Seeds from each individual plant were harvested, constituting the T1 progeny. Subsequently, seeds from the T1 progeny were individually grown following the aforementioned process. The seeds from individual lines of T1 progeny plants constituted the T2 progeny. The leaves of 5-week-old T2 progeny were collected and conducted qRT-PCR to test the overexpression of PHB-Pst-18220/PHB-Pst-18220^Δsp^, then the homozygous and high expression level seedlings of T2 progeny were utilized for the inoculation of the bacteria *Pto* DC3000. The wild-type plants were used for the control. 

### 2.9. Bacteria Pathogenicity Assay

*Pto* DC3000 was cultured overnight at 28 °C on LB agar (with 50 μg mL^−1^) and then resuspended in 10 mM MgCl_2_ (OD_600nm_ = 0.04). The bacterial cells were mixed with silwet L-77 (0.2 μL mL^−1^) and subsequently sprayed onto the leaves of 5-week-old T2-progeny plants (Homozygous line) of *A. thaliana* Col-0 with either 35S:Pst-18220 or 35S:Pst-18220^ΔSP^. The experimental plants were maintained at 22 °C (16 h:8 h; day:night) for the initial 24 h. Subsequently, the growth conditions were sustained, but with reduced humidity to 60% for the next 5 days. After 6 days of *Pto* DC3000 inoculation, the wilting black spots could be observed on the leaves of *A. thaliana*, and the percentage of necrotic area was assessed, and the leaf samples were collected for qRT-PCR testing the expression of PTI or ETI-related genes (AtPCRK1, AtPCRK2, and AtBIK1). The percentage of necrotic area (na) was calculated using the formula as follows: na (%) = [(A1/A2] × 100, where A1(necrotic area) = length × width, length as is A1 long shaft and width as is A1 short shaft. A2 is the area of the entire leaf and was defined as 100. About 48–64 leaves were used for calculating the na for each transformed *A. thaliana* genetype. Each individual treatment was replicated three times, and the whole experiment was repeated twice.

### 2.10. H_2_O_2_ Accumulation and Observation

3,3-Diaminobenzidine (DAB) staining has been demonstrated as effective in revealing the accumulation of H_2_O_2_ [27]. For this procedure, 100 mg of DAB powder (Sigma-Aldrich, TX, USA) was dissolved in 100 mL of water, and the pH of the solution was adjusted to 3.8 by adding HCl. Following 6 days of inoculation with *Pto* DC3000, the leaves of *A. thaliana* plants expressing Pst-18220/Pst-18220^ΔSP^ were harvested and submerged in the DAB solution in darkness for 8 h. Subsequently, the leaves underwent decolorization in 96% ethanol before imaging.

### 2.11. Statistical Analyses

The data for the relative expression of Pst-18220 from 0 to 264 hpi and the relative expression level of Pst-18220/Pst-18220^ΔSP^ in Pst-18220/Pst-18220^ΔSP^ overexpressing plants were examined according to Student’s *t*-test at *p* < 0.05 under the assumption of homogeneous variance. The data relating to relative transcript levels of Pst-18220-1/Pst-18220-2 in silencing wheat leaf, the hyphal length, the number of haustorial mother cells, the number of uredinium, the percentage of sites showing cell death on *N. benthamiana* infected by *Pto* DC3000, the areas of cell death on leaves of overexpressing Pst-18220/Pst-18220^ΔSP^ plants, and the relative expression levels of *AtPCRK1*, *AtPCRK2,* and *BIK1* genes in Pst-18220/Pst-18220^ΔSP^ overexpressing plants were analyzed using Duncan’s multiple range test.

## 3. Results

### 3.1. Pst-18220 Is a Pst-Specific Gene

Sixteen candidate effectors were selected from the *Puccinia striiformis* f. sp. *tritici* isolate 08/21 [17] using a developed in silico pipeline method [28]. Typically, fungal proteinaceous effectors have fewer than 300 amino acids (AA) [29]. So, an effector with 110aa is greatly accumulated in the haustoria of *Pst* during infection in wheat, which was designated as PST18220 and was chosen from the initial pool of 16 characterized candidate effectors by Petre et al. 2016 [17]. 

A TBLASTN search using PST18220 from *Pst* isolate 08/21 as a bait obtained a homologous gene in the genome of the virulent *Pst* isolate CYR32. Consequently, we obtained the full ORF length of this homolog from isolate CYR32, naming it Pst-18220. Pst-18220, with a length of 127 amino acids, lacks any known motifs, such as a nuclear localization signal, except for a 20-amino acid signal peptide (Figure 2A). To explore the genetic relationship between Pst-18220 and other rust proteins, we conducted a BLASTP search (E value < 10^−5^), finding that nearly all matching proteins belonged to *Puccinia striiformis*, except for one protein (PGTUg99_012661) from *Puccinia graminis* f. sp. *tritici* (*Pgt*). Although sequence differences exist between Pst-18220 from CYR32 and PST18220 from 08/21 (Figure 2B), they were clustered into the same large clade with other *Pst* proteins in the phylogenetic analysis (Figure 2C). This finding suggests that Pst-18220 has a low level of inter-species polymorphism within the rust family.

### 3.2. Pst-18220 Is Highly Induced at the Early Infection Stage of Wheat

To characterize the expression changes of Pst-18220 during different stages of *Pst* infestation in wheat, we employed quantitative real-time PCR (qRT-PCR) to assess the transcript levels of *Pst-18220* using the primers of PstEF1-qRT and 18220-qRT (Appendix A). The expression level of *Pst-18220* was induced upon the *Pst* germination onwheat leaves (6 hpi) (Figure 3). Subsequently, the transcript levels of Pst-18220 peaked at 72 hpi, showing an approximately 8-fold increase compared with 0 hpi (Figure 3). This peak coincided with the initiation of successful colonization of *Pst* in wheat. Following this peak, the expression of *Pst-18220* declined rapidly until 264 hpi, aligning with the time of sporulation (Figure 3). These findings suggest that *Pst-18220* is responsive to the early infection stage of *Pst*.

### 3.3. Knocking Down Pst-18220 Expression Compromised the Pathogenicity of Pst

To elucidate the role played by Pst-18220 in the pathogenicity of *Pst*, we employed the barley stripe mosaic virus (BSMV)-mediated HIGS system to silence *Pst-18220*. To specifically target the Pst-18220 sequence for silencing, we conducted a BLAST analysis and identified two segments, designated as Pst-18220VIGS-1 and Pst-18220VIGS-2 (Figure 4A), To assess the efficiency of HIGS, approximately 12 days after BSMV inoculation, all wheat samples exhibited chlorotic mosaic symptoms, except for the distinct photobleaching observed in wheat leaves inoculated with BSMV:TaPDS, indicating the successful operation of the BSMV-mediated silencing system (Figure 4B). Fewer and smaller urediniospore pustules were produced on wheat leaves inoculated with BSMV:Pst-18220-1 and BSMV:Pst-18220-2 compared with control leaves inoculated with BSMV:00 (Figure 4C). And the relative expression of Pst-18220 was down-regulated to approximately 10–50% in BSMV:Pst-18220-1 and BSMV:Pst-18220-2 wheat leaves at 24–120 h post-inoculation (hpi) (Figure 4D). Furthermore, we observed the infection process of *Pst* in wheat leaves inoculated with BSMV:Pst-18220-1 and BSMV:Pst-18220-2 (Figure 5A). The hyphal length in BSMV:Pst-18220-1 and BSMV:Pst-18220-2 inoculated leaves from 48 hpi and 120 hpi was reduced compared with BSMV:00 inoculated leaves (Figure 5B). Also, the number of haustorial mother cells from 120 hpi and the number of uredinium from 14 days post *Pst* inoculation were decreased in leaves inoculated with BSMV:Pst-18220-1 and BSMV:Pst-18220-2 in comparison with those inoculated with BSMV:00 (Figure 5C,D). These findings suggest that the down-regulation of Pst-18220 hampers the infection process of *Pst*.

### 3.4. Pst-18220 Suppressed Leaf Cell Death Mediated by Pseudomonas syringae pv. Tomato (Pto) DC3000 in Nicotiana benthamiana

The *Pto* DC3000-*N. benthamiana* interaction model is particularly relevant in the context of the typical hypersensitive response (HR) and cell death [30], which is recognized as the elicitation of ETI [31].

To further investigate whether Pst-18220 can suppress *Pto* DC3000-triggered plant ETI, we used *Agrobacterium* solutions (OD = 0.8) containing the recombinant vectors PHB-Pst-18220, PHB-Pst-18220^ΔSP^, and PHB (used as a negative control) in a solution of MES buffer. Subsequently, these solutions were infiltrated into the leaves of *N. benthamiana* using a blunt syringe (Figure 6A). After 24 h of infiltration, a solution of *Pto* DC3000 (OD = 0.01) was infiltrated in *N. benthamiana* leaves at the Agrobacterium infiltration site (Figure 6A). Following 48 h of *Pto* DC3000 infiltration, chlorotic phenotypes were not observed in all infiltrated leaf areas, except for two sites: site 7 (MgCl_2_+DC3000) and site 8 (PHB+DC3000) (Figure 6B). In particular, at site 5 (Pst-18220), it was observed that Pst-18220 can suppress the *Pto* DC3000-induced programmed cell death (PCD) in *N. benthamiana* (Figure 6B). Notably, Pst-18220 without the signal peptide (Pst-18220^ΔSP^, site 6) also inhibited the PCD triggered by *Pto* DC3000 (Figure 6B). Furthermore, the suppression of PCD by Pst-18220/Pst-18220^ΔSP^ was subsequently demonstrated using trypan blue staining (Figure 6C). The percentage of sites showing cell death in leaves expressing PHB, Pst-18220, and Pst-18220^ΔSP^ alone was almost zero. After *Pto* DC3000 infiltration, the necrosis rate of leaves expressing PHB was 100%, whereas in leaves expressing Pst-18220/Pst-18220^ΔSP^, it was significantly reduced (Figure 6D).

### 3.5. Pst-18220 Promotes Pto DC3000 Bacteria Growth in Arabidopsis

To investigate whether *Pst-18220* impairs plant immunity, we generated transgenic *Arabidopsis* lines overexpressing Pst-18220/Pst-18220^ΔSP^ through Agrobacterium-mediated transformation. Homozygous T2 progeny plants were confirmed by qRT-PCR, and the expression level of Pst-18220/Pst-18220^ΔSP^ was greatly increased more than 12-fold compared with wild-type Col-0 plants (Figure 7A). Additionally, we observed that the leaves of Pst-18220/Pst-18220^ΔSP^ -expressing plants exhibited more wilting black spots than those of the wild-type Col-0 following *Pto*DC3000 infection (Figure 7B). Furthermore, the Pst-18220/Pst-18220^ΔSP^-expressing plants display larger lesion areas compared with wild-type plants (Figure 7C,D). Concurrently, the necrotic area in transgenic plants was larger than in wild-type plants (Figure 7E). The expressions of plant immunity-related genes, namely *AtPCRK1*, *AtPCRK2*, and *AtBIK1*, which are involved in PTI or both PTI and ETI, were significantly reduced in transgenic plants (Figure 7F–H). These findings indicated that overexpression of Pst-18220/Pst-18220^ΔSP^ could reduce the immunity response of *Arabidopsis* to *Pto* DC3000 infection.

## 4. Discussion

To successfully colonize host plants, plant pathogens secrete effectors to suppress the plant defense response [32]. Numerous effectors have been identified in *Pst*, all of which contribute to the suppression of the plant defense response [10,11,15,33]. While, some effectors without any known motif play a crucial role in repressing the plant defense response. Similarly, our study reveals that Pst-18220, lacking any known domains or an abundance of specific amino acids, can compromise plant immunity.

The interaction between *N. benthamiana* and *Pto DC3000*, resulting in cell death, is considered the elicitation of ETI [31]. An effector of *Pst*, PEC6, enhances the activation of ETI induced by *Pto DC3000* infection in *N. benthamiana* [34]. However, in this study, *Pst-18220* is able to suppress the ETI triggered by *Pto DC3000* in *N. benthamiana*. The opposing effects on plant ETI between the two effectors indicate diverse mechanisms among effectors in response to plant immunity.

ROS acts as a key defense and signaling molecule during plant-pathogen interactions, and it is induced in both plant PTI and ETI [4]. Pathogen-induced ROS is involved in limiting pathogen infection, generating cell death, and signaling the transduction of defense-related processes [35,36]. Tissue necrosis caused by ROS serves opposite functions in plants encountering different pathogens. For biotrophic pathogens, duo to nutritional purposes, these pathogens prefer living tissue, while ROS-triggered necrosis leads to cell death, which triggers host resistance to biotrophs [37]. In contrast, ROS-triggered necrosis is in favor of necrotrophs, as they prefer dead cells, enhancing host susceptibility [37]. Here, as a hemibiotroph, 6 days post inoculation of *Pto* DC3000 had already entered the necrotic parasitic stage; the stable overexpression of *Pst-18220* in *Arabidopsis thaliana* led to accumulation of H_2_O_2_ and an increase in *Pto* DC3000 infection. Thus, we speculate that *Pst-18220* suppresses plant immunity in response to *Pto* DC3000 by enhancing ROS molecules and enabling growth for this bacteria. As a member of receptor-like cytoplasmic kinases (RLCKs), PCRKI increased callose deposition in *Arabidopsis thaliana* after *Pseudomonas syringae* pv. *maculicola* infection [38]. Another well-reported RLCK in *Arabidopsis thaliana* is AtBIK1, closely associated with plants PTI and ETI. This kinase directly activates Ca^2+^-permeable channel OSCA1.3 in the guard cell, controlling stomatal closure upon PAMP treatment [39]. Also, AtBIK1 activates RBOHD into ETI-associated ROS production in Arabidopsis [40]. In this study, the expression levels of *AtBIK1*, *AtPCRK1*, and its functionally redundant homologous gene AtPCRK2 in Pst-18220/Pst-18220^ΔSP^ overexpressing plants were significantly reduced compared with the wild type. Therefore, it is reasonable to speculate that Pst-18220 represses plant PTI and ETI by suppressing the expression of PTI and ETI-related genes, leading to susceptibility upon pathogen infection. This also indicates that, albeit with distinct amplitudes and dynamics, PTI and ETI could eventually converge into similar downstream responses, among many other defense responses, providing robust defense responses to *Pst* attack.

As an obligate biotrophic fungus, *Pst* grows exclusively within its plant hosts and lacks an efficient and reliable system for genetic transformation. This limitation impedes the functional verification of numerous genes, including those encoding effectors [41]. Currently, HIGS technology is commonly employed for the identification and functional research of rust effectors [33,42,43]. In this study, the silencing of *Pst-18220* led to a decrease in the number of uredinium produced by *Pst*, which suggests that Pst-18220 is a crucial virulence factor in *Pst*. Furthermore, the function of Pst-18220 was predominantly elucidated through overexpression in plants. Pst-18220 was stably overexpressed in *A. thaliana* and transiently overexpressed in *N. benthamiana*. Transient overexpression in *N. benthamiana* is a fast and convenient method for verifying the effector function of *Pst* in repressing plant immunity [17]. In this context, Pst-18220 was demonstrated to suppress cell death induced by *Pto* DC3000 in *N. benthamiana*; this is similar to an effector Pst18363 that suppressed Pst322-triggered cell death [44]. These results highlight the potential ability of Pst-18220 to interfere with the plant defense response.

## 5. Conclusions

In summary, based on the presented data, we have demonstrated that Pst-18220 plays a crucial role in *Pst* pathogenicity by suppressing both plant PTI and ETI, which give us more information about the effector without any known motif. These evidences indicate that silencing effectors without sequence features confer resistance to wheat stripe rust, which could be employed in future development of disease-resistant wheat varieties. Further studies will focus on exploring which target Pst-18220 interacts with in plant cells, which could help us clearly understand the regulation mechanism of Pst-18220 on plant immunity.

## Figures and Tables

**Figure 1 biomolecules-14-01092-f001:**
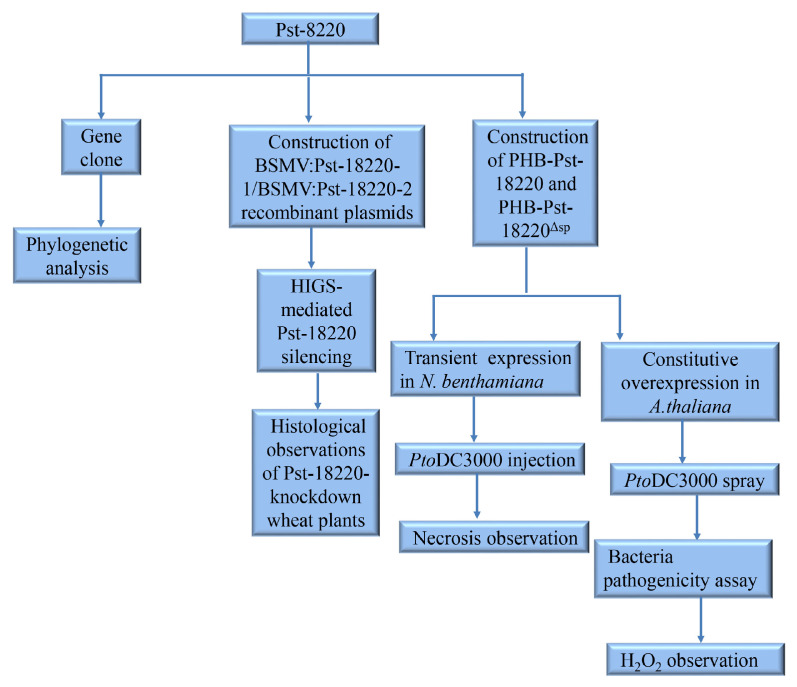
A flowchart to illustrate the experimental design.

**Figure 2 biomolecules-14-01092-f002:**
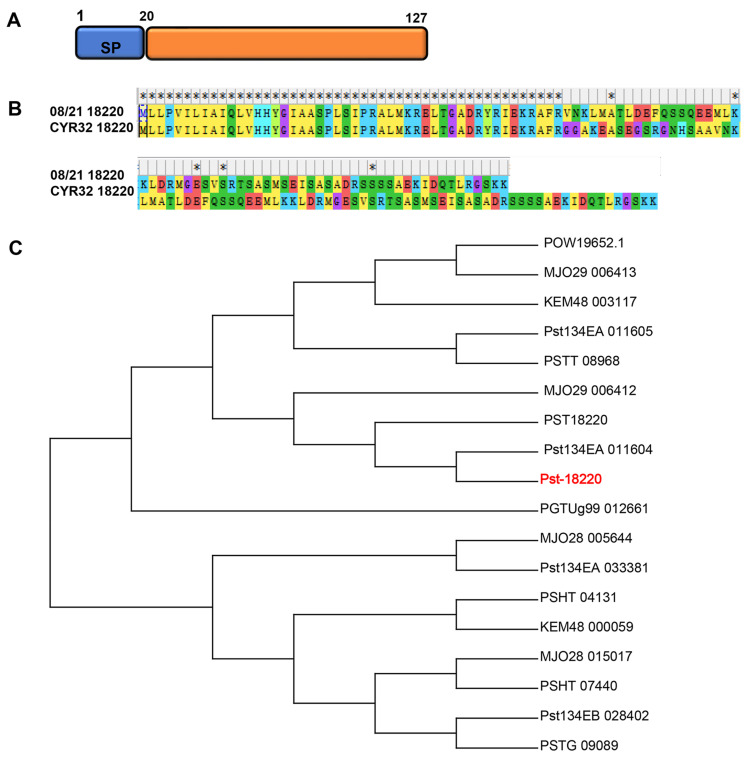
Sequences analysis of Pst-18220. (**A**) The illustration of Pst-18220. (**B**) Alignment of Pst-18220 with PST18220. * means the same amino acids. (**C**) Phylogeny of Pst-18220 orthologs. The phylogenetic tree was constructed using the minimum evolution method.

**Figure 3 biomolecules-14-01092-f003:**
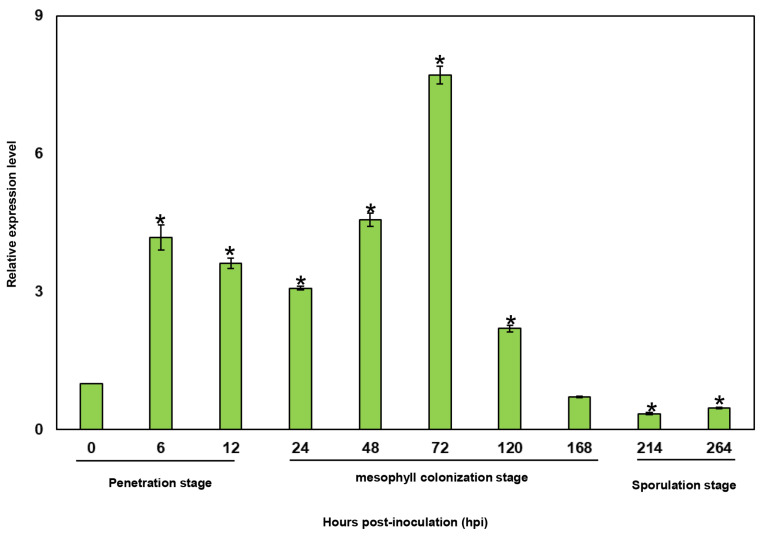
Transcript profiles of *Pst-18220* during *Pst* infection. The error bars indicate standard errors. Student’s *t*-test (*p* < 0.05) indicates significant differences in the mean *Pst-18220* relative expression level between each time point and 0 h post-inoculation (hpi) from three independent biological replications, and each biological replication includes three technical replicates. * means the significant differences between a certain time point and 0 hpi.

**Figure 4 biomolecules-14-01092-f004:**
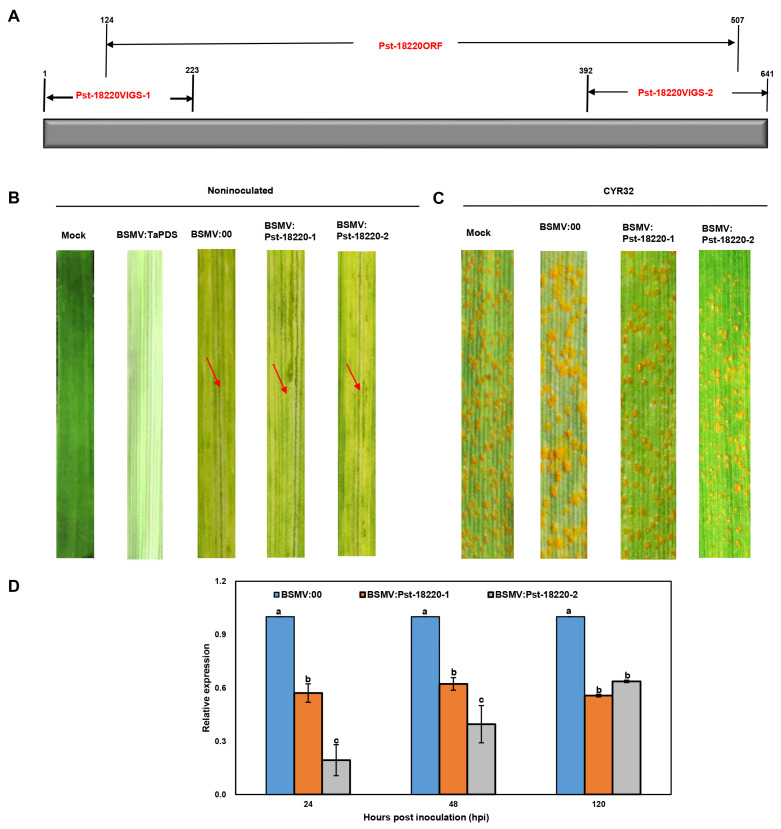
HIGS-mediated silencing of Pst-18220 in wheat. (**A**) illustration of fragment for Pst-18220 silence. (**B**) Mild chlorotic mosaic symptoms were observed on fourth leaves inoculated with BSMV:Pst-18220-1/BSMV:Pst-18220-2 at 12 days post-inoculation (dpi). Arrows indicate the mosaic symptoms. (**C**) Disease phenotypes were observed at 14 dpi on fourth leaves of wheat plants that were inoculated with virulent *Puccinia striiformis* f. sp. *tritici* (*Pst*) isolate CYR32. (**D**) Relative transcript levels of Pst-18220-1 and Pst-18220-2 in silencing wheat leaf after inoculation with CYR32. The values are the means of three independent biological replicates, and each biological replication includes three technical replicates. The error bars indicate standard errors. Analysis of variance (ANOVA) was conducted by Duncan’s multiple range test (*p* < 0.05); *PstEF* was used to normalize RNA levels in *Pst*. The means of the Pst-18220-1/Pst-18220-2 expression levels do not differ significantly if they contain at least one common lowercase letter for each treatment.

**Figure 5 biomolecules-14-01092-f005:**
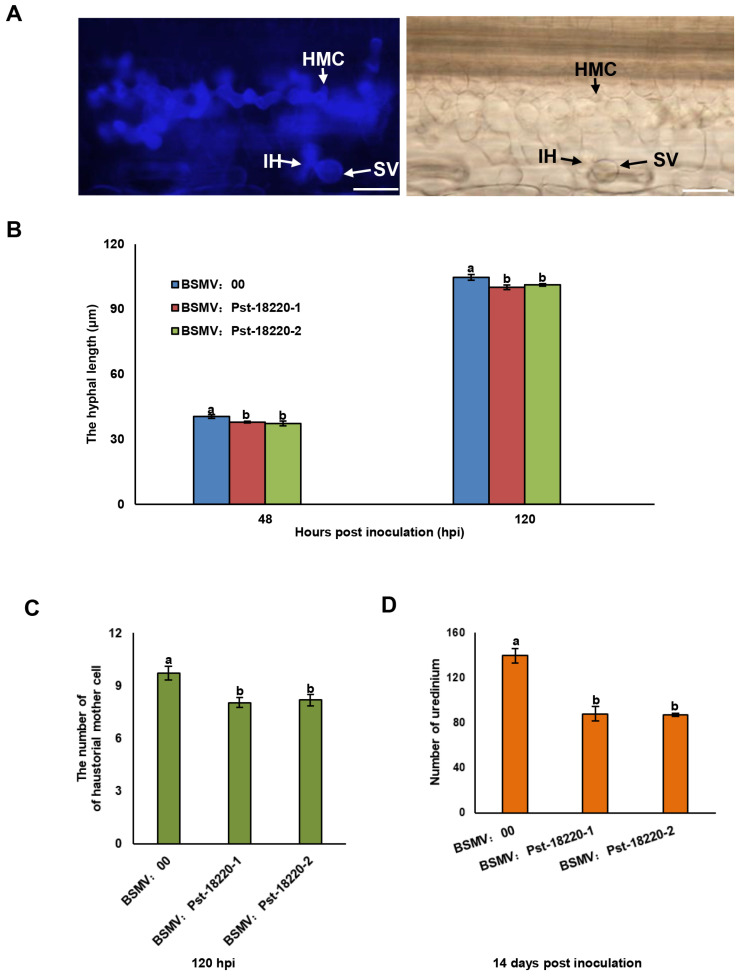
*Pst* development in HIGS-mediated *Pst-18220* silencing leaves in wheat (**A**) The observation of *Pst* infection under fluorescence microscopy at 120 hpi. Left panels: fluorescence, right panels: bright. HMC, haustorial mother cell; IH, initial hyphae; SV, substomatal vesicle. Bars, 50 μm. The development of *Pst* infection in Pst-18220 silencing leaves exhibiting in (**B**) the hyphal length, (**C**) the number of haustorial mother cell and (**D**) the number of uredinium. Hpi: hours post inoculation. The values are the means from 20–50 infection sites. The error bars indicate standard errors. Analysis of variance (ANOVA) was conducted by Duncan’s multiple range test (*p* < 0.05). The means do not differ significantly if they contain at least one common lowercase letter for each treatment.

**Figure 6 biomolecules-14-01092-f006:**
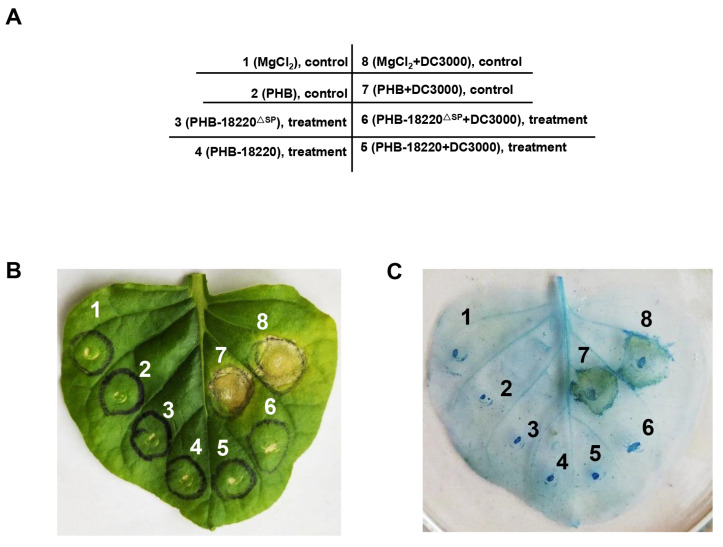
Pst-18220 suppresses *Pseudomonas syringae* pv. *Tomato (Pto)* DC3000—induced plant immunity. (**A**) The table of different treatments on *Nicotiana benthamiana* (*N. benthamiana*). (**B**) The phenotype of *N. benthamiana* upon *Pto* DC3000 infection. (**C**) The observation of trypan blue staining. (**D**) Percentage of sites showing cell death on *N. benthamiana* from different treatments infected by *Pto* DC3000. The values are the means of three independent biological replicates, and each biological replication includes three technical replicates. The error bars indicate standard errors. Analysis of variance (ANOVA) was conducted by Duncan’s multiple range test (*p* < 0.05); the means of the percentage of cell death do not differ significantly if they contain at least one common lowercase letter for each treatment.

**Figure 7 biomolecules-14-01092-f007:**
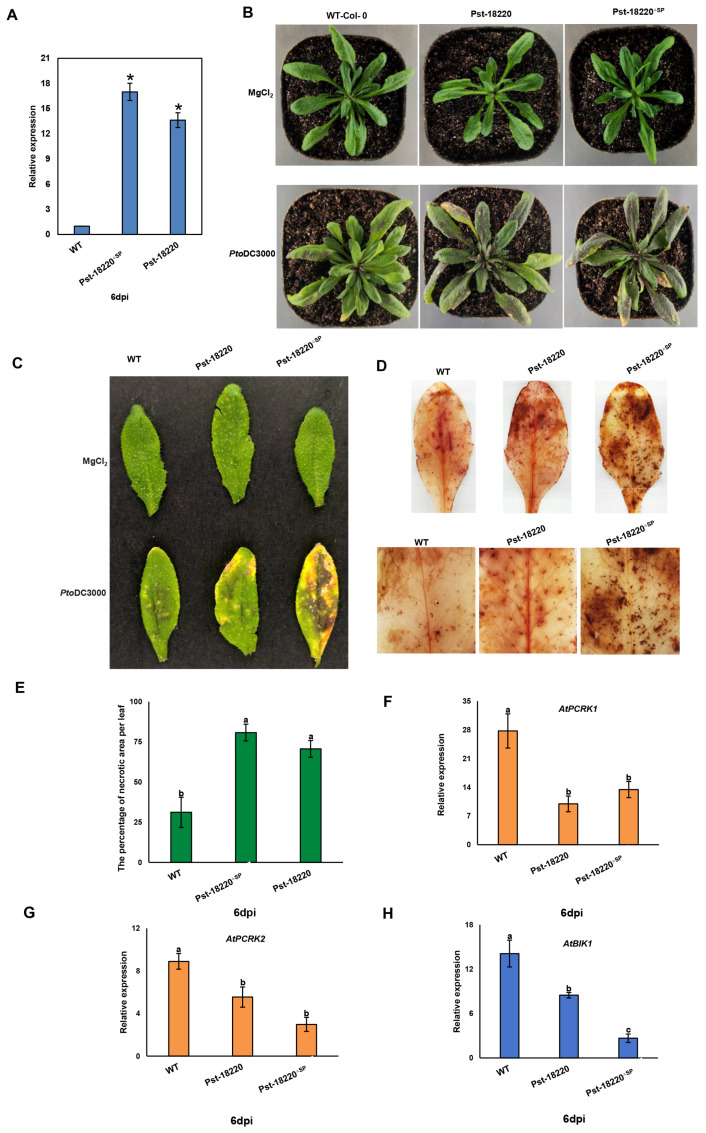
Pst-18220 overexpressing plants compromise the immunity of *Arabidopsis* infected by *Pseudomonas syringae* pv. *tomato* (*Pto*) DC3000. (**A**) The relative expression level of Pst-18220/Pst-18220^ΔSP^ in Pst-18220/Pst-18220^ΔSP^ overexpressing plants. The elongation factor *Pst-EF1* was used as reference genes. * Student’s *t*-test (*p* < 0.05) indicates significant differences in the mean relative expression level between Pst-18220/Pst-18220^ΔSP^ and wild-type Col-0 at 6 days post-inoculation (dpi) from three independent biological replications. (**B**) The phenotype on the leaves of plant overexpressing Pst-18220/Pst-18220^ΔSP^ in *Arabidopsis* after the infection of *Pto* DC3000. (**C**) The phenotype of H_2_O_2_ accumulation on Pst-18220/Pst-18220^ΔSP^ overexpressing plants by (**D**) diaminobenzidine (DAB) staining. (**E**) The areas of cell death on leaves of overexpressing Pst-18220/Pst-18220^ΔSP^ plants. (**F**–**H**) The relative expression levels of *AtPCRK1*, *AtPCRK2,* and *BIK1* genes from Pst-18220/Pst-18220^ΔSP^ overexpressing plants. *AtActin* was used to normalize RNA levels in *Arabidopsis.* The values are the means of three independent biological replicates, and each biological replication includes three technical replicates. Error bars indicate standard errors. Analysis of variance (ANOVA) for the areas was conducted by Duncan’s multiple range test (*p* < 0.05); the means of the percentage of cell death and expression levels of genes do not differ significantly if they contain at least one common lowercase letter for each treatment.

## Data Availability

The original contributions presented in the study are included in the article/Appendix A, further inquiries can be directed to the corresponding author.

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
