# Peer review of "A Putative Effector Pst-18220, from Puccinia striiformis f. sp. tritici, Participates in Rust Pathogenicity and Plant Defense Suppression"

_biomolecules, 2024, doi:10.3390/biom14091092_

Round 1
Reviewer 1 Report
Comments and Suggestions for Authors
Comment to authors
The topic of the manuscript, “A putative effector Pst-18220 from Puccinia striiformis f. sp. tritici participates in rust pathogenicity and plant defense suppression,” seems suitable for Biomolecules Journal. The manuscript is well-written, but some sections need improvement for better clarity.
Important points to be addressed:
- Pst is a biotrophic rust fungus throughout its life cycle, meaning it needs living host cells to survive and complete its life cycle. Pst has different infection structures, but all of them are part of the biotrophic way of life. Please review the text and correct accordingly.
- Particular attention should be given to the 'Material and Methods' section. The current presentation is very confusing and makes it difficult to follow the experiments undertaken. A workflow or diagram of the experimental design might help in understanding the rationale.
- Line 266: “fewer and smaller urediniospore pustules were produced on wheat leaves.” This should have been evaluated more accurately. A disease severity index should have been used to provide the percentage of symptomatic leaf area per total leaf area or the number of leaves evaluated. The presented data (photos) do not correctly represent disease severity, which is what the authors intended to show.
- Lines 271-272: The authors need to carefully review the cytological evaluation of the Pst infection process. They only mention HMC, but what about haustoria? Haustoria are the most important structures in biotrophic fungi. Without them, rust cannot complete its cycle and will not sporulate.
- Replace the expression “reactive oxygen species (ROS)” with “hydrogen peroxide (H2O2),” since the authors studied the accumulation of H2O2 and no other radicals.
- Replace the expression “cell death” with “necrosis” to avoid misleading. Since the authors evaluated “the percentage of necrotic area” caused by Pto DC3000 bacteria. Cell death (PCD) or hypersensitive response (HR) is a plant resistance response associated with biotrophic pathogens' infection (Pst), which was not evaluated.
- In the Results section - 3.4. Pst-18220 suppressed leaf cell death mediated by Pseudomonas syringae pv. tomato (Pto) 295 DC3000 in Nicotiana benthamiana: This part of the manuscript needs serious revision due to incorrect and missing information. What is the meaning of the numbers 1-8 in Figures 5B, C, and D? As far as I understand, the only sites with chlorotic phenotypes were sites 7 and 8, which is the opposite of what is written.
- Since figures should be self-explanatory, the authors need to review all figures and legends in the manuscript, with particular attention to Figure 5.
- Gene names should be written in italics. Please review the manuscript and correct accordingly.
- Whenever Pst and Pto are abbreviations of the Latin names of the pathogen species, they should also be written in italics. Please review the manuscript and correct accordingly.
- A careful review of the bibliography is necessary, as most of the italics are missing in species names and some references are not well-written or incomplete. Please ensure the references conform to the format required by the Biomolecules journal (guide for authors).
- I advise a careful revision by a senior and more experienced researcher, particular for the cytology and phytopathology issues
These and other more specific comments were made directly in the manuscript.

Reviewer 2 Report
Comments and Suggestions for Authors
This manuscript 'A putative effector Pst-18220, from Puccinia striiformis f. sp tritici, participate in rust pathogenicity and plant defense suppression' is of value to our sciences. The Pst is one of the biggest challenges to wheat production. The manuscript is good written, and the data / findings are well presented. However, i have several comments to improve it further:
Introduction: no need to mention any of your results in the introduction and instead give better focus / hypothesis on the diseases itself. The global and local wheat loses due to the diseases. Current known genes controlling the diseases!
what other control measures are present in your country?
clearly indicate your main objective/s from your study
Materials and Methods:
Materials and Methods lacking details. Alternatively, authors can support their methods by citing other refences. You are writing for others and not yourselves.
Results:
Figures caption must be well identified. Y and X access must be indicated.
Discussion: Not well explained. You must compare your results with previous similar findings.
Conclusion:
How will this study benefit our research and also our ultimate beneficiaries (farmers)? dont we have enough known markers to control this rust species? if yes, what is the benefit from identifying the Pst-18220 effector?
Format must be given more attention.
Comments on the Quality of English Language
English is fine but can be double checked again for improvement
Reviewer 3 Report
Comments and Suggestions for Authors
Comments:
1. Authors are suggested to provide background and justify why Pst-18220 was chosen.
2. Most of the abbreviations were used in this manuscript without prior definitions e.g. PTI and ETI. Author must ensure to clearly define the terms upon first use.
3. Authors mentioned in the study about the use of biological replicates without providing the information on the number of technical replicates and how variability was managed.
4. Authors are suggested to elaborate the possible pathways and molecular mechanisms through which Pst-18220 exerts its effects. In the discussion section.
5. Authors should include the potential application of current findings in developing disease-resistant wheat varieties. Authors may include a section/paragraph on how this research could translate to practical strategies for controlling wheat stripe rust.
6. Authors are suggested to improve the discussion portion. It must be lengthy/ descriptive enough to justify and compare all the findings with recent studies. In addition, limitations of the study should be enlisted and discussed.
7. Authors must re-evaluate the content of the article, ensuring that the similarity index falls within the accepted limit (below 20%).
Round 2
Reviewer 1 Report
Comments and Suggestions for Authors
Authors made the necessary correction on the manuscript.
The manuscript is accepted in the present form.
Author Response
Dear reviewer,
Thank you for your affirmation of the revised manuscripts。
Best wishes!
Junjuan Wang
Reviewer 2 Report
Comments and Suggestions for Authors
The authors have made substantial improvement on their manuscript 'A putative effector Pst-18220, from Puccinia striiformis f. sp tritici, participate in rust pathogenicity and plant defense suppression'
Comments on the Quality of English LanguageEnglish is fine just minors.
Author Response
Dear reviewer,
Thank you for your affirmation of the revised manuscripts. We have carefully checked the English again.
Best wishes!
Junjuan Wang
Reviewer 3 Report
Comments and Suggestions for Authors
Despite the revisions, significant flaws are still exist. Many of the previously raised concerns and comments have not been adequately addressed in the revised manuscript. As the persistence of these issues and the authors' inability to address them adequately, I regret to inform you that the manuscript cannot be accepted for publication in its current form. I encourage the authors to carefully consider and address the comments in any future resubmissions. Authors should limit the similarity index as well.
Author Response
Dear reviewer,
We will carefully consider the suggestions you have put forward and improve our manuscripts.
Best wishes!
Junjuan Wang